# Adherence to COVID-19 preventive measures and associated factors in Oromia regional state of Ethiopia

**Sileshi Garoma Abeya**[1]*, **Sagni Bobo Barkesa**[1], **Chala Gari Sadi**[1], **Dereje Duguma Gemeda**[1], **Fekadu Yadeta Muleta**[1], **Asebe Feyera Tolera**[2], **Dashe Negewo Ayana**[2], **Seada Ahmed Mohammed**[2], **Endale Bacha Wako**[2], **Mengistu Bekele Hurisa**[2], **Dereje Abdena Bayisa**[2], **Mirgisa Kaba Sarbesa**[3], **Eliyas Yosuf Yesuf**[4], **Asebe Amenu Tufa**[5]

1 Federal Ministry of Health, Addis Ababa, Ethiopia, 2 Oromia Region Health Bureau, Addis Ababa, Ethiopia, 3 Addis Ababa University, Addis Ababa, Ethiopia, 4 Jimma University, Addis Ababa, Ethiopia, 5 Ethiopian Public Health Institute, Addis Ababa, Ethiopia

* garomaabe@gmail.com

## Abstract

### Background

Adherence to preventive measures of Coronavirus disease 2019 (COVID-19) was among the means to tackle the transmission of the virus. However, reluctance to implement the recommended preventive measures has been reported to be a major problem everywhere including Oromia Regional State.

### Purpose

This research was aimed to assess the level of adherence to COVID-19 preventive measures and associated factors in the study area.

### Participants and methods

Community based cross-sectional study was conducted. Sample of 2751 adults aged ≥ 18 years were used for the quantitative study. Also, 20 FGDs and 30 KIIs were conducted in the qualitative approach. The collected data were entered into Epi info version 7.2.0.1 and analyzed using STATA 15. The qualitative data were entered into NVivo version 12 for its organization. Bivariate and multivariable binary logistic regression analyses were conducted to determine the association between the study variables. Odds Ratio with its 95%CI was calculated and P- Value < 0.05 was used as a cut off points to declare the significance.

### Results

The level of adherence to COVID-19 preventive measure was 8.3. Age [AOR, 4.00; 95% CI: 1.50, 10.45], Illiterate AOR, 0.38; 95% CI: 0.15, 0.93], read and write [AOR, 0.26; 95% CI: 0.10, 0.72], attended primary [AOR, 0.30; 95% CI: 0.13, 0.70], occupation (AOR; 95% CI: 0.29, 0.96] and knowledge [AOR, 0.20; 95% CI: 0.01, 0.11] were factors associated with level of adherence to COVID-19 preventive measures. Political context, unemployment,

**Data Availability Statement:** The data is available and attached as a supplementary compressed file using both STATA and SPSS spread sheets.

**Funding:** This research was conducted as a routine ministry activities during COVID-19 campaign with out a dedicated fund.

**Competing interests:** No competing interests declared.

livelihoods, and social events were mentioned as reasons for the poor adherence to COVID-19 preventive measures.

## Conclusions

The overall level of adherence to COVID-19 preventive measures in the study area was low. Age, level of education, occupation, and knowledge were factors associated with level of adherence to COVID-19 preventive measures. Activities to increase the adherence to COVID-19 preventive measures should be implemented by the concerned bodies.

## Background

COVID-19 which was reported in late December 2019 from China (Wuhan) is one of the shocking pandemics for humans [1]. The disease was declared as the sixth public health emergency of international concern [2]. Hence, this outbreak constitutes a public health risk through the international spread of the disease and requires a coordinated international response [2].

The COVID-19 pandemic reached sub-Saharan Africa by the end of February 2020 after it was declared a Public Health Emergency of international Concern by the World Health Organization (WHO) on 30, January 2020 [2]. With high levels of poverty and generally fragile health systems, sub-Saharan Africa including Ethiopia have been facing a complex regional COVID-19 epidemic and could also become a difficult task to control the virus reservoir, from where COVID-19 may be reintroduced to other regions that might have achieved control [3].

The Federal Ministry of Health of Ethiopia confirmed a COVID-19 case in Addis Ababa on March 13/2020 [4]. Generally, as of March 13, 2021, COVID-19 affected globally, over 119.7 million confirmed cases and 2.6 million deaths [5]. In Africa, over 4 million confirmed cases and over 107 thousand deaths have been reported. After the first case appeared on March 13/ 2020 in Ethiopia, the number of cases and deaths raised to +172,571and +2510, respectively in its first-year anniversary [5, 6].

Considering its pandemicity and absence of effective treatment, authorities across the globe have designed various mitigation strategies to combat the spread of COVID-19 [7]. Accordingly, to limit the transmission, the WHO recommends minimizing contact between infected and non-infected persons, early detection and isolation of cases, and general personal and collective hygiene measures [6, 8, 9]. As part of these measures, the use of face masks, hand washing, physical distancing, cough etiquette and avoidance of crowded places are recommended [9].

Although adherence to preventive measures is the only means to tackle the virus. Reluctance to do so has been reported to be a major problem everywhere [7]. Also, community's risk perception and poor adherence to COVID-19 mitigation measures remains a major problem. A significant proportion of communities did not perceive the virus as a risk for health [10]. People also think that it originated from a laboratory, and mostly causes mild symptoms, and affects the elderly [10, 11]. On the other way, there is no effective treatment for the COVID-19 infection till now. Henceforth, adherence to COVID-19 preventive and control measures is the only option to stop its spread and minimize its disastrous impact on developing nations like Ethiopia so that the knowledge and behavior changes are pillars to engage with preventive measures [1].

In the study conducted in Vietnamese people showed that 88.2%, 99.5%, 94.9%, and 97.4% of the participants adhered to the physical distancing rule, wear a face mask, cover mouth and nose during coughing/sneezing and wash hands regularly with water and soap, respectively [1]. Similarly, in Iran 95.4%, 93%, and 80% of the participants adhered to hand washing with soap and water, avoiding crowded places, clean hands with other disinfectants, respectively and showed a good adherence to COVID-19 preventive measures [12].

In contrast to the above findings, the adherence level of COVD-19 preventive measures in Africa was low [13]. In a knowledge, attitude and practice assessment survey of Africa, only 12.3% of the study participants adhered to the suggested COVID-19 preventive measures, although some preventive measures like avoidance of handshaking, eating uncooked food, gatherings, and frequent hand washing were implemented by 81.4%, 77.2%, 69.9%, and 65.8%, respectively. Similarly, in Egypt and Nigeria, only 36% of the participants implemented all the recommended measures despite most (96%) practiced self-isolation and physical distancing [13]. In Uganda, only 29% were adherent to all the preventive measures [14]. In contrast, in Mozambique, the use of wearing face masks, regular hand washing and cough hygiene all reached compliance rates of over 90% [15].

According to the study conducted in Gondar city of Ethiopia, nearly half of the study participants (48.96%) had poor adherence to COVID-19 preventive measures [7]. Among the preventive strategies, hand washing was the most (73.84%) preventive measures practiced by the respondents, while a significant proportion (67.58%) failed to use a face mask [7]. On the other hand, in Derashe district of southern Ethiopia, better results of the selected preventive measures were seen; avoidance of handshaking and gatherings with many people were reported by 81.4% and 69.9% of the study participants, respectively [1]. In the same study, wearing face masks (20.5%) and staying at home (22.8%) were the least practiced preventive measures [1]. Moreover, none of the participants implemented the entire recommended COVID-19 preventive measures [1].

In a study of North Shoa zone of Ethiopia, the overall adherence level of the community to the recommended safety measures of COVID-19 was 44.1% [2]. Thus, this study was conducted to assess the level of adherence to COVID-19 preventive measures and associated factors in Oromia regional state.

## Methods and materials

### Study area

This study was conducted in ten selected zones and towns of Oromia Regional State. Oromia is one of the largest and most populous regions in Ethiopia with an estimated population of 39,074, 846. The region is divided into 20 zones, 19 Administrative towns and 333 districts. The dominant livelihood of the residents of the region is Agrarianism, Agro-pastoralism and Pastoralism. The region borders the Somali Regional State to the east; the Amhara region, the Afar region and the Benishangul Gumuz region to the north; South Sudan and the State of Benishangul/Gumuz to the West; Gambella region, SNNP region and Kenya to the south. In the region, there are four specialized referral hospitals, five university hospitals, 34 general hospitals, 47 primary hospitals, 1410 health centers and 7090 health posts.

### Study design and period

A community-based cross-sectional design was conducted using a mixed method of quantitative and qualitative approaches from September 2020 to March 2021.

## Population

All adults living in the Oromia Regional State during the study period were the source population, while all adults living in the selected households of Oromia region during the study period were the study population.

**Inclusion and exclusion criteria.**   Men and women aged 18 years and above who have been residing in the area for at least six months were included in the study, while those who were critically sick, hearing difficulties and unable to communicate during the data collection time were excluded from the study.

**Sample size determination.**   *Quantitative*. The sample required to compare the adherence of COVID-19 preventive measures to urban population in comparison to rural population was calculated using a formula to compare and test the difference between the two population proportions using comparative study design. In using this formula, the following assumptions were considered: the proportion of adherence to COVID-19 preventive measures among urban population is 50% in the absence of previous study. Under null hypothesis, where the two proportions are not differing from each other, the maximum tolerable level of difference $(P_1 - P_2)$ between the two population proportions of 0.05 was considered. Accordingly, $P_1 = 0.50$ and $P_2 = 0.45$. were taken. And their average (pooled proportion) was obtained by adding the two proportions and dividing by two [16]. The following assumptions were also considered in calculating the sample size;

$n_1$ = an independent sample size in urban population

$n_2$ = an independent sample size in rural population

$P_1$ = proportion of respondents who adhered to of COVID-19 preventive measures an urban population,

$P_2$ = proportion of respondents who adhered to of COVID-19 preventive measures in rural population,

P = an average of the proportions COVID-19 preventive measures acceptance in two groups

$Z_{\alpha/2}$ = the corresponding value of confidence coefficient at alpha level of 0.05 that is 1.96

$Z_{\beta}$ = the corresponding value of power at beta level of 0.2 that is 0.84

With these assumptions, considering the scenario where the alternate hypothesis is true and the proportions are significantly different and the general formulae is given as follows.

$$n = \frac{\left\{Z_{1-\alpha/2}\sqrt{2\overline{P}(1-\overline{P})} + Z_{1-\beta}\sqrt{[P_1(1-P_1) + P_2(1-P_2)]}\right\}^2}{(P_1 - P_2)^2}.$$

[16].

By replacing the corresponding values for the symbols in the formulae and having design effect of two and adding for the possible non response rate, a total of 2851 respondents were obtained where 1426 from urban areas and another 1426 respondents from rural areas (districts) were selected.

*Qualitative*. The intention of the study was to explore the perception, knowledge, attitudes and practices of people towards the COVID-19 preventive measures. The required information for this purpose was not only acquired through survey but also through qualitative methods. This approach was applied based on the assumption that it allows to triangulate the method and data. That is, in addition to the collection of quantifiable information using survey method, the qualitative method helps to explore the lived experiences of the study participants in the context of COVID-19. In this regard, the qualitative method supplements the quantitative findings with evidence generation. More specifically, the qualitative method mainly aimed at addressing the "why" people in the study area adhere/not adhere to the preventive practices

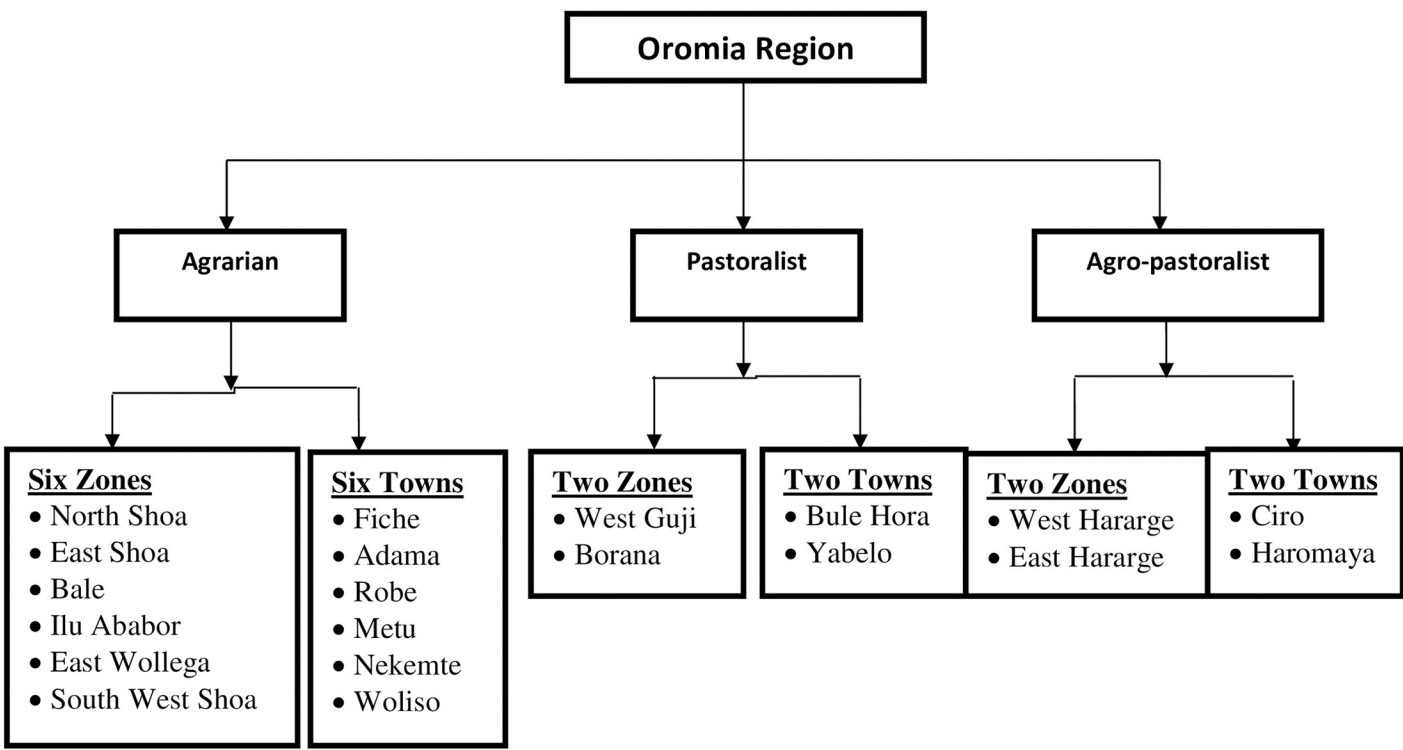

**Fig 1. Schematic presentations of the sampling procedures to select the zones and towns, Oromia region, September 2020 to March 2021.**

of COVID-19. Hence, Key Informant Interview (KIIs) among healthcare managers and Focused Group Discussion (FGDs) among community members were used as methods of the qualitative data collection.

**Sampling procedures.** *Qualitative*. The region was categorized into three clusters; namely, agrarians, semi-pastoralists, and pastoralists. The main reason to use this method was based on the fact that the region is heterogeneous with regard to economic, cultural, geographic and climatic conditions. From each geographic area zones and towns were randomly selected and three districts per zone and three sub-cities per town were also randomly selected to have the participants from the households. After identifying households in the respective districts and sub-cities participants were randomly selected to be included in the study. Using this method, the residents have equal and independent chances of being enrolled in the study (Fig 1).

Moreover, based on the aforementioned livelihood clusters, zones and towns the desired eligible sample was allocated proportionally. Using population to size proportionate methods, the required sample size was determined in each study site. Then, using systematic sample by calculating interval (total HH population of the area/sample size) preferably the household heads or the available eligible were selected and included in the study (Table 1).

*Qualitative*. About 30 key informant interviews were conducted in the study with zonal health administration officials and Woreda administration officers. Thorough discussions were made with them on adherence of COVID-19 prevention methods. Also, 20 FGDs with adult women and men in all sample districts were held.

Participants for the qualitative survey were identified, screened for eligibility, and selected by purposive sampling method by the coordinators from each selected site. The criteria for the selection include their detailed knowledge of COVID-19 preventive measures. The

**Table 1. Proportion of sample size allocated to zones and towns of the study area, Oromia region, September 2020 to March 2021.**

| Description | Urban sample size 1426 | | | | |
|---|---|---|---|---|---|
| | Rural sample size = 1426 | | | | |
| Zones /Towns | Zones pop$^n$ | Zone pop$^n$ Minus Urban | Sample per zone | Town pop$^n$ | Sample per/town |
| **Agrarian** | | | **703** | | **1125** |
| North Showa/Fiche | 1690403 | 1645269 | 141 | 45134 | 70 |
| East Shoa/Adama | 1615178 | 1229941 | 105 | 385237 | 600 |
| Bale /Robe | 1886779 | 1813919 | 155 | 72860 | 113 |
| Ilu-Ababbora/Metu | 991,257 | 943105 | 81 | 48152 | 75 |
| East Wollega /Nekemte | 1634387 | 1510903 | 129 | 123484 | 192 |
| South West Shoa / Woliso | 1126028 | 1077684 | 92 | 48344 | 75 |
| **Pastoralists** | | | **703** | | **1125** |
| West Guji/Bule Hora | 1523137 | 1465246 | 125 | 57891 | 90 |
| Borena /Yabello | 566406 | 539682 | 46 | 26724 | 45 |
| **Agro pastoralist** | | | **172** | | **135** |
| West Harargie /Chiro | 2667000 | 2611725 | 223 | 55275 | 86 |
| East Harargie/ Haromaya | 3882018 | 3831573 | 328 | 50445 | 79 |

participants include policymakers, service providers and service users from each of the participating areas. Research collaborators in the six sites who were all healthcare practitioners working in the tertiary health facilities supported the recruitment of policymakers and service providers in the area. The service providers helped in the recruitment of the service users who were getting healthcare services during the study period (Fig 2).

## Data collection

The quantitative part of the study involves the collection of quantifiable and measurable data on the implementation of COVID-19 preventive measures endorsed by the government. In this regard, the preparation of the questionnaire was based on conceptual framework of the study and similar previous research work to answer the objectives. The questionnaire was first prepared in English, and then translated into Amharic and *Afaan Oromoo* for data collection and back to English by different people to ensure its consistency. Health professionals having diploma and above were recruited based on their previous experiences of data collection and interest for data collection. Data were collected by face-to-face interview from the eligible. One participant was randomly selected from each household in case there were two or more respondents to prevent intra-household correlation.

The qualitative component was intended to explore the barriers being experienced by the people on COVID-19 preventive measures. It tried to explore the level of preparedness of Oromia Region to respond to such pandemic and examine the available policies and action plans that are currently in place. In doing so, FGDs and KIIs guides were prepared separately in English and translated into *Afaan Oromoo* to address the specific research questions. During data collection, the interviewees were asked about their regular living situations as well as previous conditions to see the implementation of COVID-19 preventive measures. The interviews and discussions were managed by face-to-face in Amharic and *Afaan Oromo* languages using trained data collectors (Sociologists and medical anthropologists). Before starting the actual data collections, pre-test was conducted and comments were acquired on the tools. Comments were also obtained from colleagues, advisors and other concerned bodies.

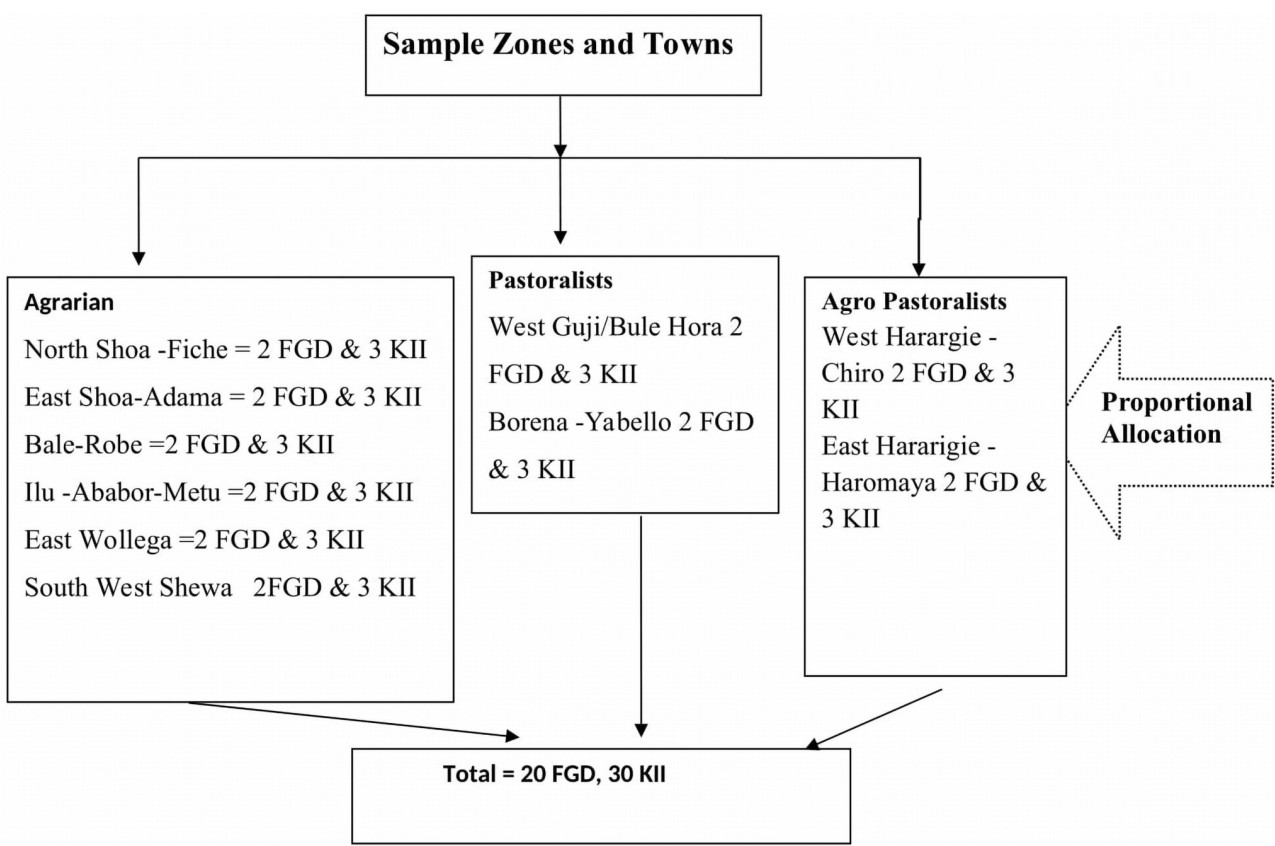

**Fig 2. Schematic presentation of sampling procedure for qualitative methods, Oromia region, September 2020 to March 2021.**

For the qualitative data collection, coordinators identified the potential participants of the study, scheduled the dates and time of the planned FGDs and KIIs after obtaining informed verbal and written consent. The participants were informed that the interview would be recorded during the consent process. Interviews and discussions were then being conducted face-to- face by trained interviewers and discussion facilitators. Indeed, the interviewers and discussion facilitators have expertise in conducting FGDs and KIIs and were working with members of the central research team. The study participants were informed about the purpose of the study and were invited to participate in the interview and discussions, which lasted for approximately 40–60 minutes each. The interviews and discussions were conducted using facilitators guide designed specifically for each of the intended groups. After the interview, the audio recordings were transcribed verbatim.

## Quality assurance

To enhance the quality of the instruments of data collection, pre-testing of the questionnaire was undertaken prior to data collection. In addition, a-three-day training was given for data collectors and supervisors concerning the objective, the tools, methodology, and ethical issues. During the data collection period, the collected data were checked for completeness and consistency by the supervisors and principal investigators. Moreover, each supervisor was given his/

her own household enumerators and data collectors and reoriented them during each day before data collection. They also supervised them by crosschecking the registered households and administered questionnaire for its completeness. Before starting data entry, unique codes were given to each questionnaire. Missing values and outliers were checked using frequency tabulations, residual plotting and managed accordingly. Data was edited and checked manually by hand for checking completeness both during collection and entering into data entry templates.

**Data management.** Data was entered into Epi info version 7.2.0.1, data entry template and exported to STATA 15 software for analysis. Missing values and outliers were checked by frequency tabulations. Randomly selected 5% of the data set was double entered to check the accuracy and similarities based on the questionnaire's identification numbers. Any decision or change used on the data set was clearly documented for further explanation of unexpected errors that may happen at the end of the day. In addition, check for item and unit-missing values, outliers for accuracy, causes of outliers were considered and determined.

**Data analysis.** The quantitative data was analyzed using STATA 15 software. Descriptive statistical analysis such as frequency, percentage, proportions with 95% CI, mean and standard deviation were used. The associations between level of adherence to COVID-19 preventive measures and independent variables were modeled using binary logistic regression analysis. Simple logistic regression analysis was used to assess the existence crude relationship between independent variables and level of adherence to COVID-19 preventive measures. At this level the candidate selected independent variables for multiple regression analysis at P-value < 0.25 significance level. Multiple logistic regressions were applied to estimate the adjusted effects of independent variables on the level of adherence to COVID-19 preventive measures. The regression model was developed using forward stepwise approach. The odds of being adhered to COVID-19 preventive measures were estimated using odds ratio within 95% confidence intervals. At this level, the significance of associations was declared at p-value of 0.05.

The final fitted model was assessed for assumptions like normality of continuous variables using histogram and normal curve, multicollinearity between independent variables using Variance Inflation Factor (VIF) and goodness of fit using Hosmer and Lemishow test. Moreover, the model ability to correctly classify those subjects who experience outcome of interest and those who do not was assessed using Receiver Operating Characteristics (ROC) curve. Findings were presented on frequency tables, graphs and discussed accordingly.

The qualitative data analysis was begun with the work of transcription, translation and theme development during data collection. Initially the KII and FGD were transcribed and translated. Then a workshop was prepared to develop themes by reading all translated data. The data was then entered into NVivo version 12 for its organization and management.

## Operational definitions and measurements

**Level of adherence.** Adherence towards prevention and control measures for COVID-19 was computed from the response category of the preventive measures endorsed by the government (hand washing, using a facemask, keeping physical distance, not travel to a crowded place, home stay, and not travel to anyplace during the pandemic) regularly practiced during 14 days before data collection time. The score was computed from those who properly practiced. Those respondents who scored 95% and above were labeled to have "Good adherence to COVID-19 preventive measures" and otherwise [7].

## Ethics consideration

Ethical approval was obtained from the ethical review board of Oromia Regional State Health Bureau. Permission letters were secured from Regional and Zonal Health Offices and shared

with the randomly selected health care facilities and community administrators. Written and verbal consent were obtained from participants.

Approval and permission were sought from the concerned bodies for the study. The ethical review was undertaken by all project and investigators ensured standard processes (dignity, autonomy, informed consent, confidentiality, anonymity, ability to adhere to protocol) and data security are maintained. Voluntary and informed participation, confidentiality and safety of participants constituted key principles of researcher respondent interaction. Informed verbal and written consents were obtained from residents, service users, service providers and policymakers prior to their enrolment in the study. The study was conducted according to the Helsinki declarations on ethical principles for medical research involving human subjects. Finally, the collected data was stored in a separate computer and kept confidentially. On completion of the study, both the quantitative and transcribed data were stored in password-protected computers/laptops and only the core research team has access to the data.

## Results

### Socio-demographic characteristics of the study participants

About 2851 sample were planned of which 2724 were participated in the study making the response rate of 95.5%. The mean (±SD) age of the respondents was 33.30 (±11.34) years ranged from 18 to 82 years. More than one third (36.8%) of the participants were within the range from 26 to 35 years. The majority of the respondents, 1512 (73.6%), were from agrarian, while 203(9.9%) were from agro-pastoralists cluster areas. Of the respondents 1503 (56.7%) were urban residents and more than half, 1333(51.4%), were females. Most, 1818 (68.1%), of the respondents were married during data collection period. Regarding their religion, most, (41%), of them were Orthodox Christian followers. The majority, 2170 (79.9%), of the respondents were Oromo by their ethnicity and 847 (31.2%) were farmers or pastoralists by their occupation; whereas 393 (14.3%) were students. About quarter of the respondents, 681 (25.1%), attended secondary school education and about one in ten, 306 (11.3%), could read and write. The estimated annual income of the respondents ranged from 23 to 14,773 USD with the median (±IQR) of 227 (±14,750) USD (Table 2).

### Knowledge about COVID-19

The majority of the respondents, 2525 (91.6%) have heard about COVID-19, but only 61.3% believe the existence of COVID-19 in their area. Moreover, less than one in ten, 258 (9.36%), of the respondents believes as COVID-19 is a killer disease (Fig 3).

The qualitative data also shows that people have information about the disease. However, due to lack of COVID-19 cases and morbidity, some believe that their area is free from the disease. For instance, a FGD discussant in Bale Zone Dinsho district said,

*I believe this disease does not exist in our area. In our neighbor, people also believe in a similar way. The communities have awareness on corona and maintain physical distancing. All schools were closed except 8th grade. Children were not allowed to play together. We wear face masks when we went to market and in transport to strictly protect ourselves.*

**A 37 years male**

Moreover, it was found that there was a difference between rural and urban communities with regard to the kowledge of COVID-19 including its preventive measures. A male FGD

**Table 2. Socio-demographic characteristics of the respondents, Oromia region, Ethiopia, September 2020 to March 2021.**

| Variables | Response Category | Number | Percent |
|---|---|---|---|
| **Cluster of Respondent (n = 2055)** | Agrarian | 1512 | 73.6 |
| | Agro-Pastoralists | 203 | 9.9 |
| | Pastoralist | 340 | 16.5 |
| **Residence of Respondent (n = 2651)** | Urban/Town | 1503 | 56.70 |
| | Rural/Woreda | 1148 | 43.30 |
| **Sex of Respondent (n = 2591)** | Male | 1258 | 48.6 |
| | Female | 1333 | 51.4 |
| **Age** | 18–25 yrs. | 725 | 28.1 |
| | 26–35 yrs. | 949 | 36.8 |
| | 36–45 yrs. | 603 | 23.4 |
| | 46–55 yrs. | 171 | 6.6 |
| | $\geq$ 55 yrs. | 133 | 5.2 |
| **Marital Status of Respondent (n = 2670)** | Single | 706 | 26.4 |
| | Married | 1818 | 68.1 |
| | Widowed/Divorced/separated | 146 | 5.5 |
| **Religion of Respondent (n = 2723)** | Orthodox | 1111 | 41 |
| | Muslim | 982 | 36 |
| | Protestant | 555 | 20 |
| | Others* | 75 | 3 |
| **Ethnicity of Respondent (n = 2717)** | Oromo | 2170 | 79.9 |
| | Amhara | 344 | 12.7 |
| | Tigre | 32 | 1.2 |
| | Others** | 171 | 6.3 |
| **Occupation of Respondent (n = 2724)** | Farmer or pastoralist | 847 | 31.2 |
| | Merchant | 632 | 23.3 |
| | Student | 393 | 14.5 |
| | Gov./NGO worker | 408 | 15.0 |
| | Others *** | 433 | 16.0 |
| **Level of Education for Respondent (n = 2709)** | Illiterate | 498 | 18.4 |
| | Read and write | 306 | 11.3 |
| | Primary | 590 | 21.8 |
| | Secondary | 681 | 25.1 |
| | Colleges and above | 634 | 23.4 |
| **Estimated annual Income** | Less or equal to 227 USD | 932 | 50.4 |
| | 228–562 USD | 384 | 20.8 |
| | 563–1,136 USD | 297 | 16.1 |
| | 1,137–2,273 USD | 185 | 10.0 |
| | $\geq$ 2274 USD | 52 | 2.8 |

NB: Others include

* Catholic and Wakefata

** Sidama, Wolayita and Gurage

*** Work in private organization, house maid, and daily laborer.

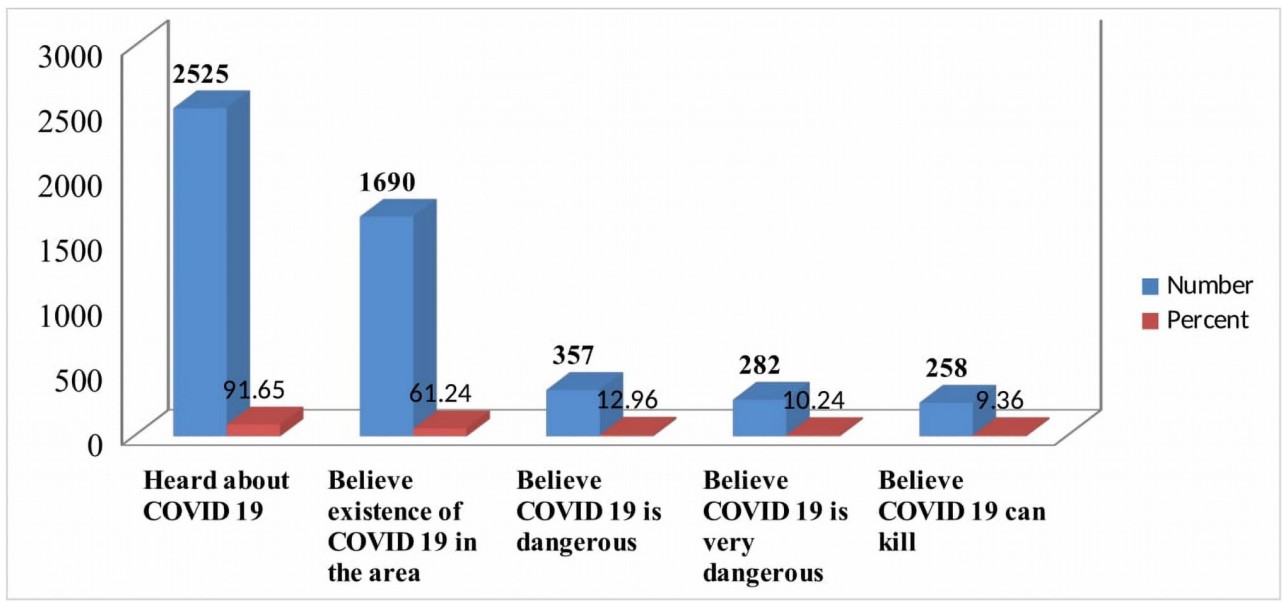

**Fig 3. Perception of respondents about COVID-19, Oromia region, Ethiopia, September 2020 to March 2021.**

held in East Wollega illustrated that people in urban area have more access to sources of information and are more knowledgeable than rural people. A participant in the FGD said,

*There is a difference between people in rural and urban area of our community with regard to knowledge. The people in urban area have more knowledge than the rural area in having information from different source but they are reluctant in implementing prevention measure. They relate COVID-19 with politics so that awareness creation campaign becomes ignored. This factor by itself is also one of the hindering factors of application of prevention measure than lack of knowledge.*

**A 45 years male**

Regarding media and other source exposure, more than one in six, (63.09%), of the respondents have got information about COVID-19 on radio and from the health workers conveyed for about 43.88% of the respondents. The least proportion (14.4%) heard information about the COVID-19 from their close friends (Fig 4).

It was also reported that people have multiple sources of information about the disease regardless of age difference. For example, a female key informant in Nekemte city said, "W*e have enough information from different sources like mainstream media. Awareness creation was also made previously so we know ways of its transmission and prevention. In my area, everybody including children and elders know COVID-19.*" **35 years Female**

The participants in the qualitative method also reported that social media such as Facebook, telegram, WhatsApp and Twitter were the main sources of information about COVID-19. A key informant in Borana Zone Health Office illustrated saying, *"We have got the information from social Medias of global and national individual and institutional actors informing that the disease is highly spreading worldwide."* **A 28 years male**

Of the respondents most, (67.7%) and (63.6%), have information about the protection measures and symptoms, respectively. The least proportion (23.8%), informed about the risks/complications of COVID-19 (Fig 5).

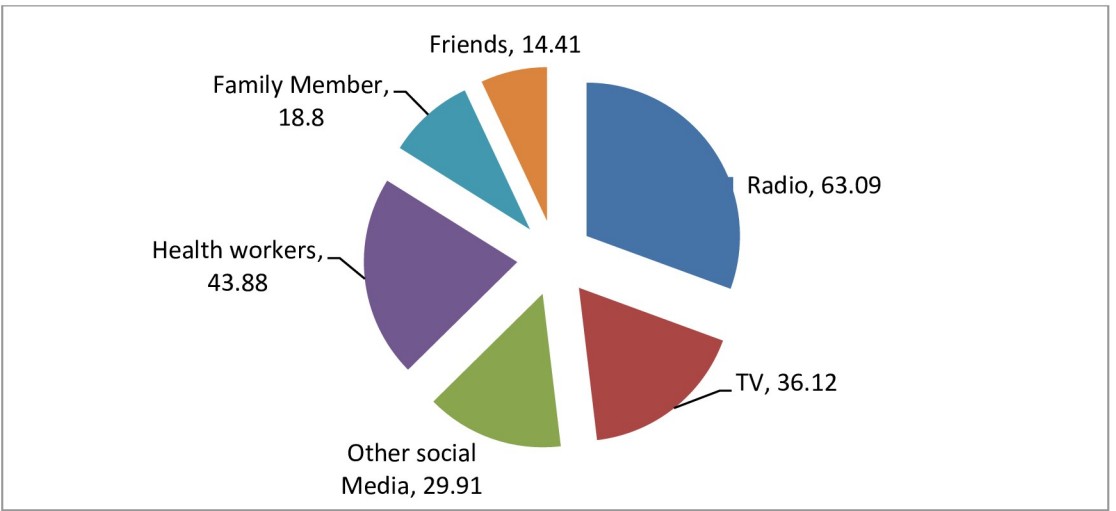

**Fig 4. Percentage of source of information, Oromia region, Ethiopia, September 2020 to March 2021.**

Regarding information about the preventive measures of COVID-19, the vast majority, (80.9%), of the participants reported to have information about regular hand washing using hand rub or soap and water, while (7.5%) of them didn't know the prevention measures of COVID-19 during the data collection time (Fig 6).

Regarding the perception of the study participants on the transmission of COVID-19, the majority (65.6%), perceived that the droplets spitted from infected person is the most transmission way of COVID-19 to other people. About 9% perceived the transmission way by sexual intercourse. Also, about **2.3%** of the respondents didn't perceive anything about the transmission way for COVID-19 (Fig 7).

The study participants were also asked about the symptoms experienced by a person infected with COVID-19 when sick. Accordingly, about eight in ten, (79.4%) and (75.93%),

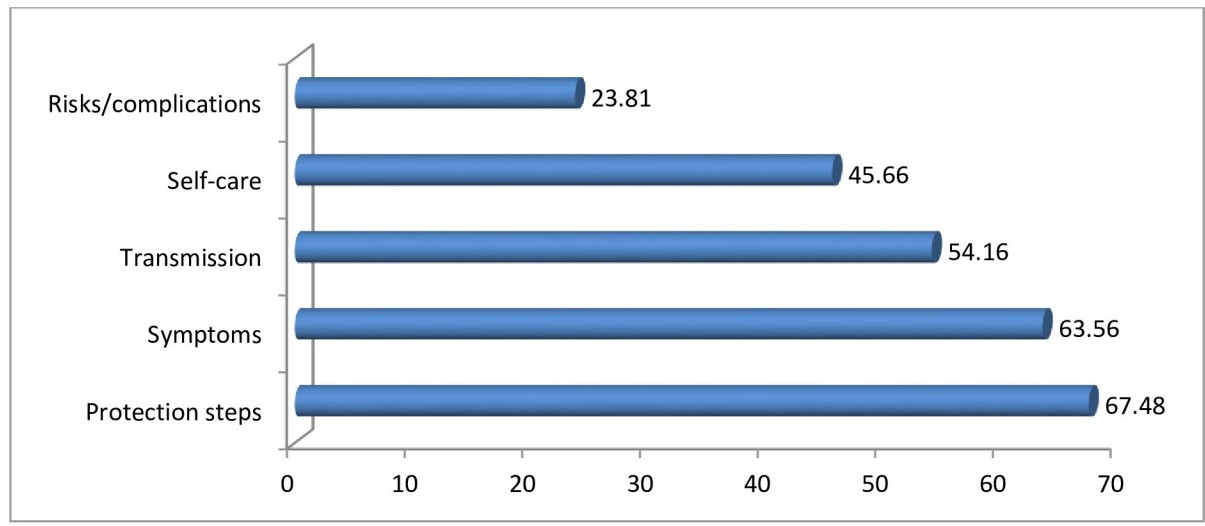

**Fig 5. Percentage on kinds of information on COVID-19, Oromia region, Ethiopia, September 2020 to March 2021.** NB: Percentage may not add 100% as multiple responses were possible.

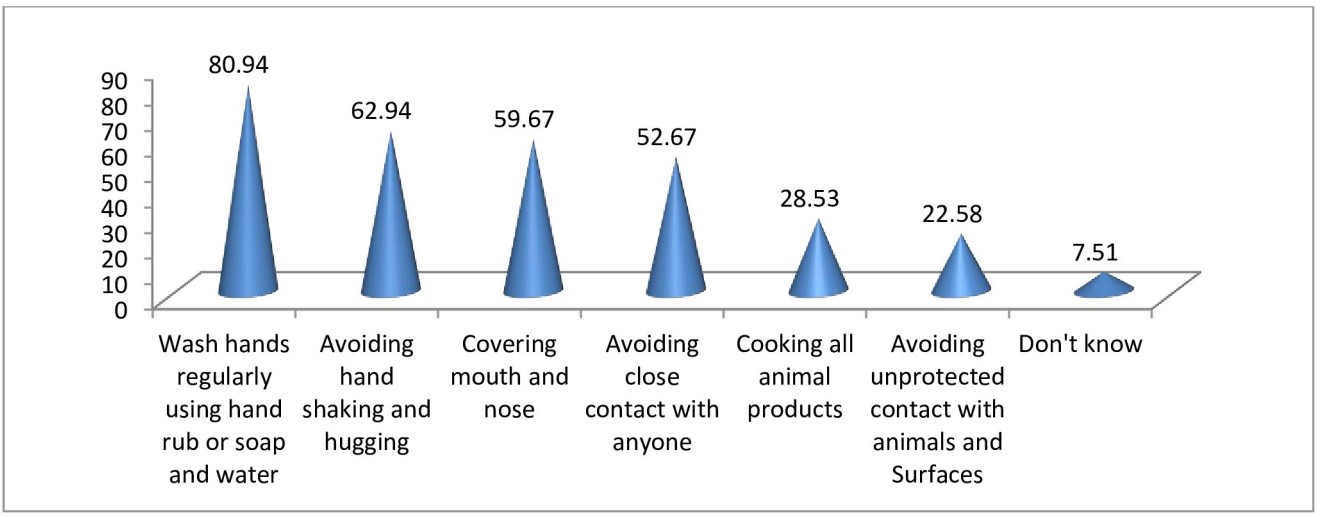

**Fig 6. Percentage of information among respondents about COVID-19 preventive measures, Oromia region, Ethiopia, September 2020 to March 2021.** NB: Percentage may not add 100% as multiple responses were possible.

mentioned cough and fever, respectively, as the main symptom of COVID-19 when a person gets sick. Whereas, nearly 1% of the participants responded as there is no symptom from the COVID-19 infected person when sick (Fig 8).

The composite knowledge score was calculated from the above knowledge related variables and accordingly 1606 (58.4%) have good level of knowledge and 1139 (41.5%) were labeled to have poor knowledge about COVID-19 and its preventive measures. Moreover, the stratified

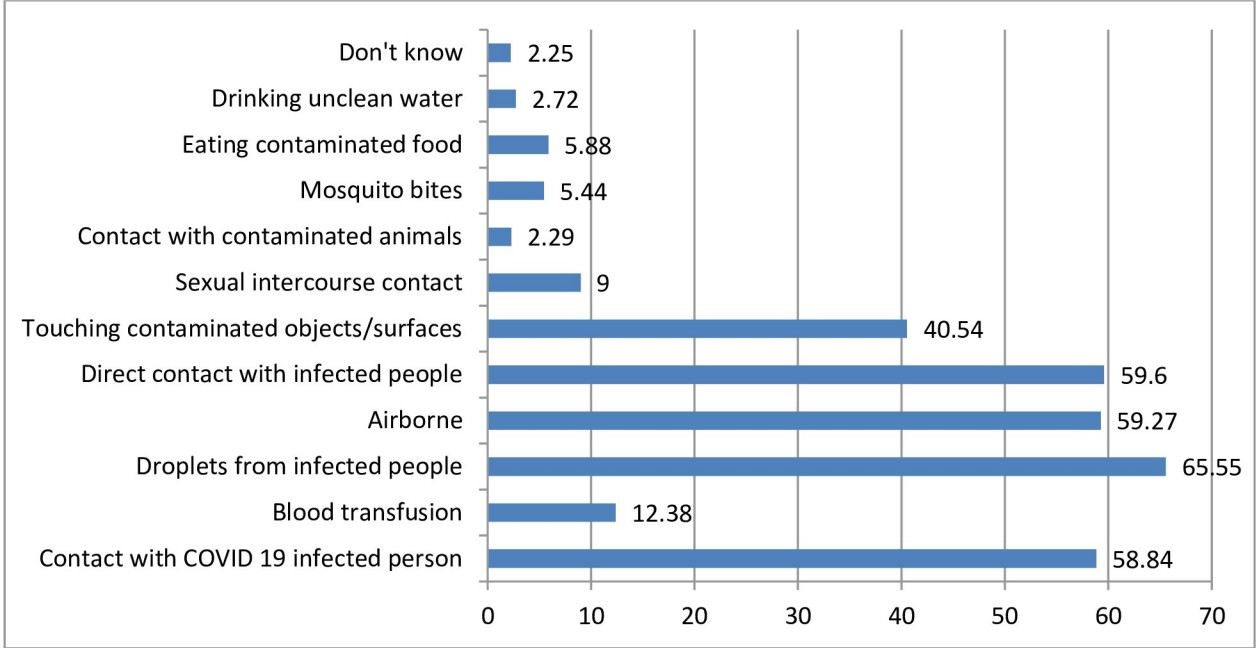

**Fig 7. Percentage of respondent's perception on the transmission of COVID-19, Oromia region, Ethiopia, September 2020 to March 2021.** NB: Percentage may not add 100% as multiple responses were possible.

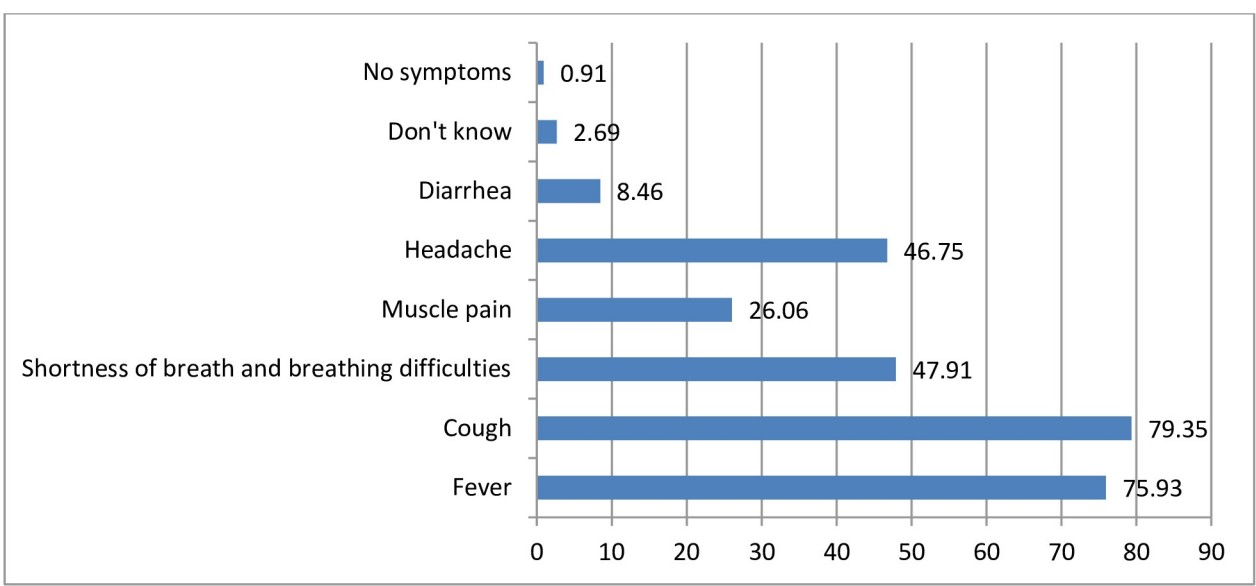

**Fig 8. Percentage of the symptoms of COVID-19 mentioned by the respondents, Oromia region, Ethiopia, September 2020 to March 2021.** NB: Percentage may not add 100% as multiple responses were possible.

analysis according to area of residence showed 63.0% of urban/town and 52.3% of rural/ district residents had good level of knowledge, respectively ($X^2$ = 30.8, P< 0.001).

## Attitudes towards COVID-19 preventive measures

The attitudes of the study participants were explored using Likert scale measures. Accordingly, 2355 (86.6%) confirmed that COVID-19 is a killer disease, whereas 223 (8.2%) of them disagreed to its severity. On the other hand, the majority of the study respondents, 2378 (87.6) agreed as COVID-19 is a preventable disease. The majority of the study respondents, 1220 (44.9%), disagreed to the government responsibility of implementing the preventive measures of COVID-19, while most, 2229 (82.2%), of the respondents agreed to the communities' responsibility in implementing COVID-19 preventive measures. Similarly, about nine in ten, 242 7(89.8%), of them agreed to the individual's responsibility to apply all the preventive measures against COVID-19. Conversely, one in three, 871 (32.2%), of the respondents have favorable attitudes towards COVID-19 Preventive measures (Table 3).

The qualitative methods have explored various attitudes towards COVID-19. Accordingly, it was found that some people believe COVID-19 affects people differently. In this regard, there was an assumption in the community believing that the disease does not affect young people. A male FGD participant conducted from Nekemte city said,

> There is misconception that Coronavirus have no serious effect on younger people especially for less than 40 years old. So, when you ask young people why they are not wearing facemask they say the virus is not risky for us, rather let the older ones wear." If wearing is mandatory, they wear masks on their beard. Generally, the reasons for not implementation of prevention measure are that I didn't have seen Coronavirus.

**A 46 years male**

**Table 3. Attitudes of the respondents towards COVID-19 preventive measures, Oromia region, Ethiopia, September 2020 to March 2021.**

| Variables | Response Category | Number | Percentage |
|---|---|---|---|
| COVID-19 is a killer disease (n = 2718) | Agree | 2355 | 86.6 |
| | Neutral | 140 | 5.2 |
| | Disagree | 223 | 8.2 |
| COVID-19 is preventable (n = 2716) | Agree | 2378 | 87.56 |
| | Neutral | 198 | 7.29 |
| | Disagree | 140 | 5.15 |
| Government is responsible for implementing the preventive measures of COVID-19 (n = 2716) | Agree | 1213 | 44.7 |
| | Neutral | 283 | 10.4 |
| | Disagree | 1220 | 44.9 |
| Community is responsible for implementing preventive measures of COVID-19 (n = 2713) | Agree | 2229 | 82.2 |
| | Neutral | 166 | 6.1 |
| | Disagree | 318 | 11.7 |
| Individuals are responsible to apply all the preventive measures of COVID-19 (n = 2703) | Agree | 2427 | 89.8 |
| | Neutral | 127 | 4.7 |
| | Disagree | 149 | 5.5 |
| Attitude score | Unfavorable | 1838 | 67.8 |
| | Favorable | 871 | 32.2 |

Moreover, there were people who could not believe even in the existence of the disease. Due to lack of confirmed or COVID-19 morbidity, some people considered that the disease does not exist in their area. For instance, an FGD discussant in South West Shoa mentioned that he does not believe the existence of the disease as he didn't see any pain on COVID-19 infected people. To put in his own words,

*There is hospital in this town called Luke hospital that serve as COVID-19 treatment center so we hear that one person dead of covid-19. Firstly, government sectors was creating an awareness about all prevention measures, but daily we are seeing people discharging from isolation center and said we never feel any pain so that we realized that there is no COVID-19 around. In other word, our people deny that there was no disease called COVID-19.*

**A 30 years male**

The study participants also alleged that COVID-19 was politicized. For instance, a key informant interview in Yabello town described that "some of the community member believe that there is no disease. The government talks about it for political purpose." Another informant in Woliso town mentioned that "there is no Coronavirus rather the government is politicizing it, but we are telling the community that the virus is real and life-threatening disease and no political need behind." Similarly, another key informant who was working as health officer in Borena Zone illustrated that *"there are negative attitudes towards the disease assuming that it is political game for postponing election."*

Furthermore, some discussants and informants of the study related COVID-19 more to spirits than to a real disease and considered it as a wrath of God and evil spirit. A key informant in Bule Hora town mentioned as follows:

*Some of the community members believe that there is no disease even preached at some religious institutions. The town closed three Protestant churches following this wrong act against*

*COVID-19. The religious members consider it as an evil spirit or Satan's act on human being and nothing to do.*

**A 41 years male**

## Practice and adherence of the COVID-19 preventive measures

The study participants were asked about the measures that were taken at least once since the start of COVID-19 pandemic. Accordingly, the vast majority, 2131 (84.0%), have tried washing hands regularly using hand rub or soap and water. Similarly, 70.8% and 83.1% of the study participants have practiced avoiding hand shaking or hugging and covering their mouth and nose, respectively. Moreover, 85.4% of the study participants practiced at least one of the preventive measures endorsed by the government. However, about 4.4% of them did nothing to prevent COVID-19. When the adherence level of the preventive measures was computed for the regular and usual practices for 14 days prior to data collection time, 8.3% (95%CI: 7.7, 8.9) used to have the practices for about 95% and above and labeled to have a good level of adherences to COVID-19 preventive measures and otherwise (Table 4).

When asked for the reasons for not or poorly adhering to COVID-19 preventive measures a month prior to a survey, the vast majority, (82%), kept quiet or not responded to the questions and the insignificant number (4%) believed in their own religion for not to have COVID-19 infection. Also, about 6% perceived as COVID-19 is not a killer disease and no need for the frequent use of preventive measures (Fig 9).

Participants were asked whether COVID-19 preventive measures were being practiced within their family members or not. Accordingly, the vast majority (93.76%) witnessed that they have practiced in covering their mouth and nose during coughing and sneezing. Moreover, about quarter (24.83%) of the family members have practiced at least any one of the preventive measures (Fig 10).

## The difference between awareness, knowledge, attitude and adherence to COVID-19

Even though the awareness level of people was extremely high, a decreasing trend was seen across the knowledge, attitude and adherence to COVID-19 preventive measures (Fig 11).

**Table 4. Practices and adherences of study participants to COVID-19 preventive measures, Oromia region, Ethiopia, September 2020 to March 2021.**

| Preventive measures | Number | Percent | 95%CI |
|---|---|---|---|
| Wash hands regularly using hand rub or soap and water (n = 2537) | 2131 | 84.0 | 83.27, 84.73 |
| Avoiding hand shaking and hugging (2457) | 1739 | 70.8 | 69.88, 71.72 |
| Covering mouth and nose (n = 2502) | 2079 | 83.1 | 82.35, 83.85 |
| Avoiding close contact with anyone (n = 2238) | 1034 | 46.2 | 45.20, 47.20 |
| Cooking all animal products (n = 2071) | 670 | 23.4 | 22.50, 25.30 |
| Avoiding unprotected direct contact with live animals and Surfaces (n = 1981) | 302 | 15.2 | 14.39, 16.01 |
| Practice at least one of the above | 2434 | 85.4 | 84.68, 86.12 |
| Did nothing (1952) | 86 | 4.4 | 3.30, 5.50 |
| Level of Adherences (1970) | | | |
| • Poor | 1807 | 91.7 | 91.08, 92.32 |
| • Good | 163 | 8.3 | 7.55, 9.05 |

NB: Percentage may not add 100% as multiple responses were possible except for the adherence Category.

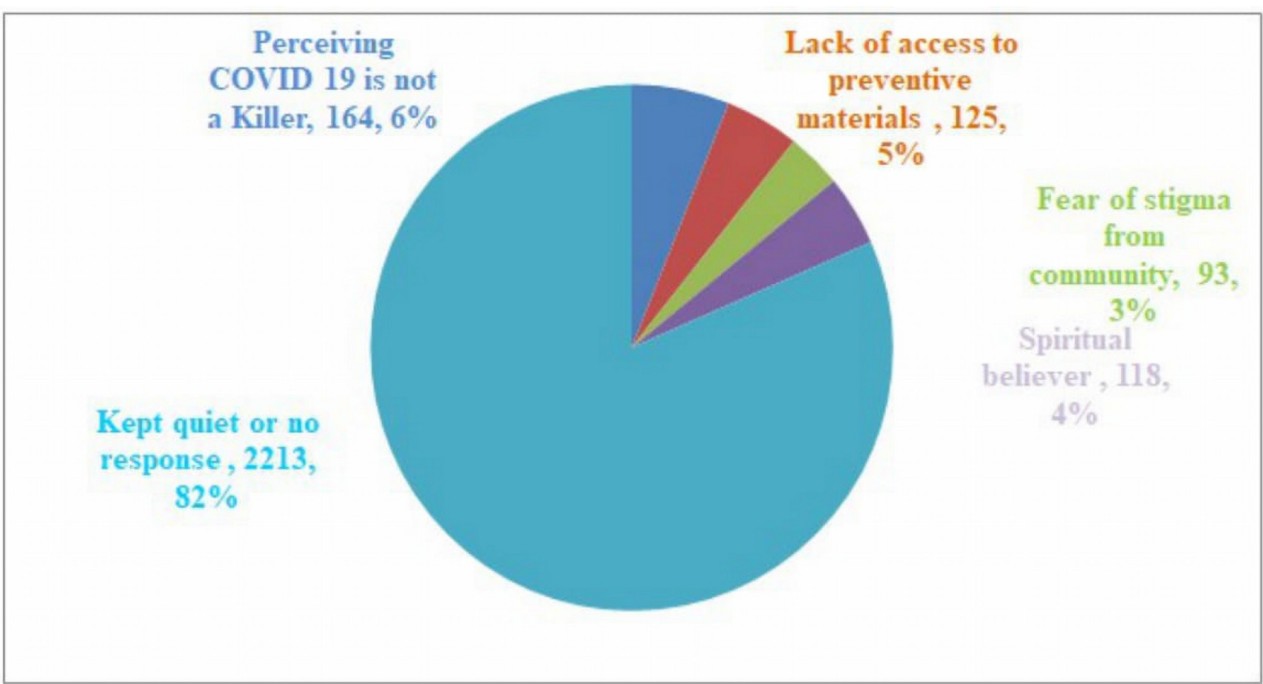

**Fig 9. Reason for not adhering to COVID-19 preventive measures among study participants, Oromia region, Ethiopia, September 2020 to March 2021.**

### Factors associated with adherence to COVID-19 preventive measures

Geographical Cluster, participant's age, Occupation, level of education, Level of knowledge, and attitudes were selected to be input into the final model (P< 0.21). All candidate variables selected by simple logistic regression analysis were subjected to multiple logistic regression models to estimate their adjusted effect on level of adherence to COVID-19 Preventive

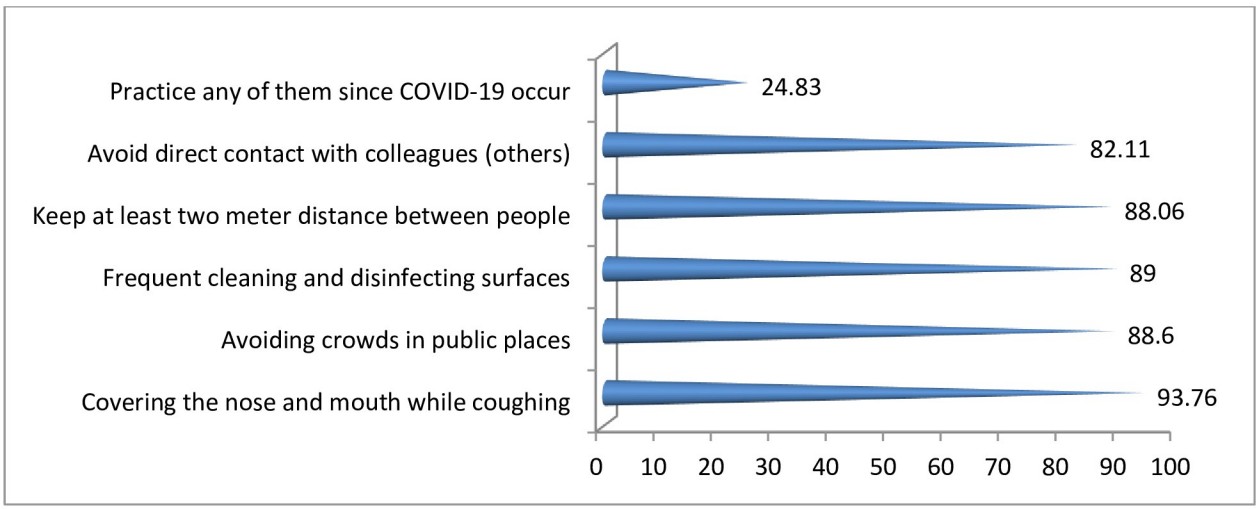

**Fig 10. Percentage of measures taken by respondents, Oromia region, Ethiopia, September 2020 to March 2021.** NB: Percentage may not add 100% as multiple responses were possible.

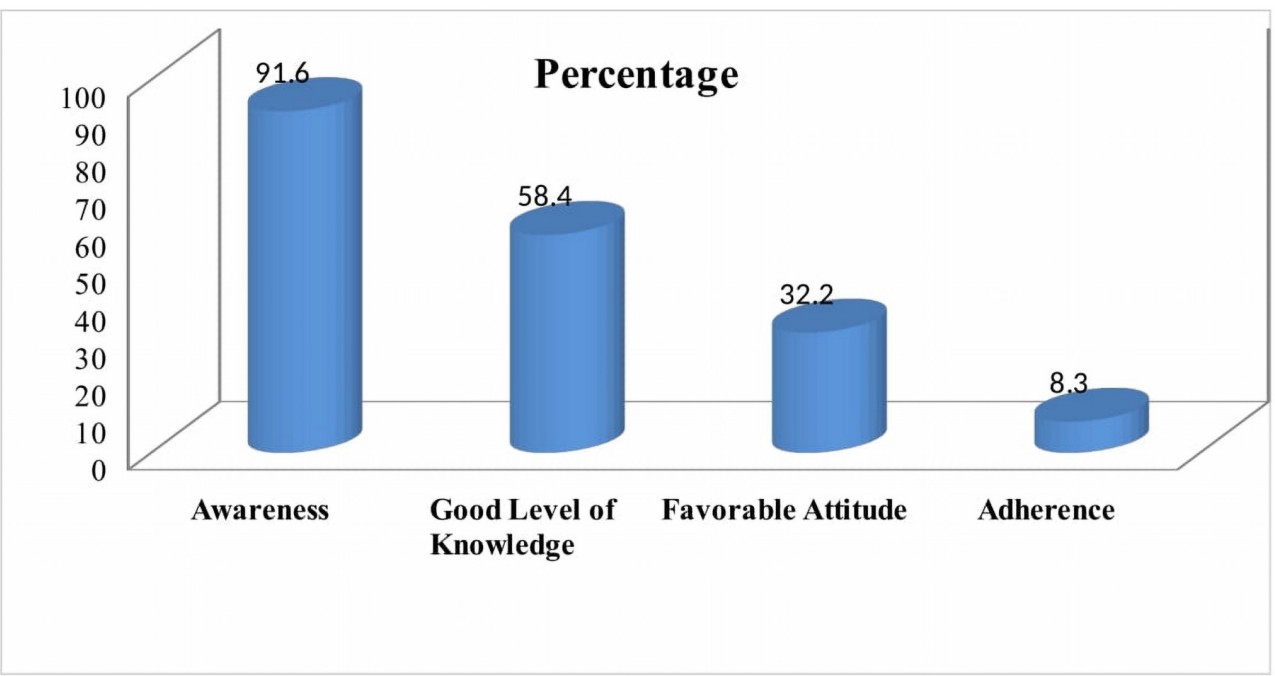

**Fig 11. Percentage differences across awareness, knowledge, attitude and adherence to COVID-19 preventive measures, Oromia region, Ethiopia, September 2020 to March 2021.**

measures. In this model, the independent effect of explanatory variables was estimated by controlling the effects of possible confounders. Accordingly, after adjusted to possible confounders, participants age, level of education and level of knowledge were found significantly associated with level of adherence to COVID-19 Preventive measures at P-value < 0.05.

Based on the current finding, being older age group of 36–45 years was four times [AOR, 4.00; 95% CI: 1.50, 10.45] more likely to have good adherence to COVID-19 preventive measures compared to those aged 18–25 Yrs. The odds of good level of adherences towards COVID-19 preventive measures increase with level of education. Accordingly, compared to the study participants who attended Colleges and above, being Illiterate [AOR, 0.38; 95% CI: 0.15, 0.93], can read and write [AOR, 0.26; 95% CI: 0.10, 0.72], and attended primary [AOR, 0.30; 95% CI: 0.13, 0.70] were less likely to have good level of adherence to VOVID 19 preventive measures. Being merchant were less (AOR; 0.62; 95% CI: 0.29, 0.96] likely to adhere to COVID-19 preventive measures compared to farmers. The study also showed that the odds of having good adherence to COVID-19 preventive measures lower [AOR, 0.20; 95% CI: 0.01, 0.11] among participants with poor level of knowledge on COVID-19 (Table 5).

The qualitative method also explored hindering factors for applying COVID-19 prevention mechanisms. For instance, socio-economic problems, lack of COVID-19 confirmed cases, low enforcement mechanisms and low level of perceiving risk were the main reasons for not practicing the prevention methods. One of the KII in Dinsho district described as follows;

*Economic problem, politics and culture can be the reasons for not practice COVID-19 prevention method. Massive meeting and rallies conducted in different place that we observed affected our community to decrease the practice of COVID-19 prevention methods.*

**A 40 years male**

**Table 5. Factors associated with level of adherence to the prevention of COVID-19 among respondents, Oromia region, Ethiopia, September 2020 to March 2021.**

| Variables | Response Category | Level of Adherence | | COR [95%CI] | AOR [95%CI] |
|---|---|---|---|---|---|
| | | Poor | Good | | |
| Cluster | Agrarian | 937 (91.0) | 99 (9.0) | 1:00 (Ref.) | 1:00 (Ref.) |
| | Agro-Pastoralists | 131 (95.6) | 12 (4.4) | 0.87 [0.46, 1.62] | 1.23 [0.48, 3.14] |
| | Pastoralist | 194 (96.8) | 6 (3.3) | **0.29 [0.13, 0.68]*** | 2.25 [0.62, 8.24] |
| Age | 18–25 Yrs | 488 (93.7) | 33 (6.3) | 1:00 (Ref.) | 1:00 (Ref.) |
| | 26–35 Yrs | 602 (92.3) | 50 (7.7) | 1.23 [0.78, 1.94] | 1.40 [0.56, 3.42] |
| | 36–45 Yrs | 379 (87.3) | 55 (12.7) | **2.15 [1.37, 3.40]**** | **4.00 [1.50, 10.45]*** |
| | 46–55 Yrs | 121 (89.0) | 15 (11.0) | 1.83 [0.97, 3.48] | 3.03 [0.91, 10.10] |
| | > 56 Yrs | 84 (92.3) | 7 (7.7) | 1.23 [0.53, 2.88] | 1.95 [0.50, 8.10] |
| Ethnicity | Oromo | 1444 (91.7) | 131 (8.3) | 1:00 (Ref.) | 1:00 (Ref.) |
| | Amhara | 212 (89.8) | 24 (10.2) | 1.25 ().79, 1.97] | |
| | Others ∞ | 129 (97.7) | 3 (2.3) | **0.26 [0.08, 0.82) *** | |
| Occupation | Farmer or pastoralist | 578 (91.5) | 54 (8.5) | 1:00 (Ref.) | 1:00 (Ref.) |
| | Merchant | 451 (94.5) | 26 (5.5) | **0.62 [0.38, 0.90]*** | **0.53 [0.29, 0.96]*** |
| | Student | 251 (92.6) | 20 (7.4) | 0.85 [0.50, 1.45] | 0.82 [0.20, 3.34] |
| | Gov./NGO worker | 249 (86.5) | 39 (13.5) | **1.68 [1.08, 2.60]*** | 0.40 [0.11, 1.11] |
| | Others∞∞ | 262 (91.9) | 23 (8.1) | 0.94 [0.56, 1.56] | 0.25 [0.10, 1.10] |
| Level of Education | Illiterate | 330 (93.0) | 25 (7.0) | **0.52 [0.32, 0.85]*** | **0.38 [0.15, 0.93]*** |
| | Read and write | 216 (94.7) | 12 (5.3) | **0.38 [0.20, 0.73]**** | **0.26 [0.10, 0.72]*** |
| | Primary | 400 (92.4) | 33 (7.6) | **0.57 [0.36, 0.89]*** | **0.30 [0.13, 0.70]*** |
| | Secondary | 433 (93.1) | 32 (6.9) | **0.51 [0.32, 0.80]**** | 0.49 [0.23, 1.03] |
| | Colleges and above | 406 (87.3) | 59 (12.7) | 1:00 (Ref.) | 1:00 (Ref.) |
| Level of knowledge | Poor | 791 (97.9) | 17 (2.1) | **0.15 [0.09, 0.25]***** | **0.20 [0.01, 0.11]***** |
| | Good | 1016 (87.4) | 146 (12.6) | 1:00 (Ref.) | 1:00 (Ref.) |
| Level of Attitude | Unfavorable | 1165 (95.7) | 92 (7.3) | **0.70 [0.50, 0.97]*** | 0.84 [0.50, 1.46] |
| | Favorable | 619 (89.8) | 70 (10.2) | 1:00 (Ref.) | 1:00 (Ref.) |

NB:

* P < 0.05;

** P < 0.01;

*** P < 0.001

Others include

∞ = Tigre, Sidama, Wolayita and Gurage

∞∞ = Work in private organization, house maid, and daily laborer.

As mentioned in the above quotation, there were political events (in support of or against the existing political system) such as rally (public meeting) and violence that brought many people together created conducive environment for the spread of the disease. These circumstances made people to be careless and avoid using prevention methods. A key informant in Bale Zone Health Office pointed out one of the incidents as follows:

*The possible challenges not to use the preventive methods were the mass grievance and violence after the death of artist Hacalu Hundessa that the community said no disease but the political actors are the virus by themselves.*

**A 34 years male**

The discussants and informants also described that there was lack of or loose law enforcement to re-enforce people in the use of the prevention methods in their area. The informant said that,

*In the beginning, law enforcement by the government had helped for proper utilization of COVID-19 prevention methods. Religious and cultural leaders are also played, major role in helping the community to proper use of COVID-19 prevention method. Later on, this law enforcement from the government declined. The people start to stop utilization of COVID-19 prevention methods. Currently, public gathering is underway without proper care in our area. Keeping social distancing and personal hygiene is not properly practiced in our zone.*

*A 51 years male*

Furthermore, lack of commitment from the side of the government itself made the rules of the prevention measures to be oversighted. Example, the key informant from Dinsho district described that the government itself did not adhere to the rules. He said, "*We advised on different preventive methods and we practiced as much as we could after attentive follow. As a political concern, we observed that there were still meetings of many people by the government during the time of corona.*" *A 28 years Female*

The study discussant and informant also mentioned absence of COVID-19 related morbidity and confirmed cases as one of the main reasons for not complying with COVID-19 prevention methods. A key informant working as zonal PHEM in Borena described *that "low morbidity, absence of sign and symptoms on those who diagnosed by laboratory has significantly decreased the fear and adherence of COVID-19 prevention method our community."* ***A 41 years male***

Economic problems were among the key factors for the non-implementation of COVID-19 prevention methods. For instance, a male FGD discussant in East Wollega zone illustrated as follows:

*Student and other living in this town have awareness about Coronavirus. But they all joking in its implementation sometimes we hear that "our priority is not Coronavirus" Young people graduated and unemployed in this town. Generally, they all know about the disease.*

It is possible to understand from the above excerpt that the **massive unemployment** in the area is reported as one of the key challenges beyond COVID-19. Lack sustainable and adequate employment for newly graduating youths created hopelessness and doesn't create fear for the disease. Another key informant in Woliso town mentioned that "most people in this town are hopeless because of unemployment. So, how can they hear what you are trying to teach them? This is another factor."

Lack of adequate source of income is also a key issue in making the prevention measures available for all. The study participant in Bale described that,

*as you know most of populations are in low income, as we talk of prevention measures most of them need money that may be difficult for some of our people, for instance, soap and alcohol, etc. need money that some of our community can't afford."*

*A 56 years male*

Moreover, some of the **livelihood activity** practiced in the study area were found not to adhere to COVID-19 prevention rules. For example, works such as farming (working in groups such as Dabo), daily laborer, and petty trade are some of the works where the

respondents were unable to comply staying at home, maintaining physical distance, and avoiding gathering. A female FGD participant in Adama district described:

*individual economic status is among the hindering factors to practice preventive measures, for example, if someone have enough money to feed his/her family, he/she can practice stay at home preventive measure but if not, one cannot practice it. Even families lack money to buy face mask and sanitizer for their children.*

**A 46 years female**

In the study area, the necessity of participation in **social events** such as *idder* also made the prevention mechanisms unrealistic. Membership and participation in societal events such as wedding, burial, and *idder* are vital for ones live in the communities. Hence, people were urged to participate in the gatherings otherwise they will be isolated and cannot survive. A female FGD discussant in Woliso district explained,

*We can't manage the number of people during death ceremony because some of our people say "I can't avoid this gathering as we are the same idder" so this issue may be one of hindering factor.*

**A 40 years female**

Another female FGD discussant in Adama town illustrated that:

*Our culture of togetherness like celebrating festivals together, funeral services and other ceremonies are hindering practices of Coronavirus preventive measures. For example, if someone not participated on funeral services of neighbors, he/she will be marginalized and disrespected in the village.*

**A 30 years male**

## Discussion

This study used a community based cross-sectional design using both quantitative and qualitative approaches. It aimed to assess the level of adherence to COVID-19 preventive measures in terms of a composite score comprising of six measures endorsed by WHO and the Government: Washing hands regularly using hand rub or soap and water (hand hygiene), Avoiding hand shaking and hugging, Covering mouth and nose (face mask use), Avoiding close contact with anyone (physical distancing), Cooking all animal products, and Avoiding unprotected direct contact with live animals and Surfaces and associated factors. In doing so, the level of Knowledge, attitude and practices of the COVID-19 preventive measures were computed.

During the study period, the majority of the participants **(91.6%)** had heard about the world COVID-19 pandemic, and the majority had heard the information on the radio. About 58.4% had a good level of knowledge from the composite score calculated from related variables. This result is lower than studies done in China 90% [17]. This may be because developing countries use social media less than developed countries and minimize disruption caused by the Coronavirus. However, it is similar to the findings from a study conducted in bi-national African countries (Egypt and Nigeria) showed for 61.6% had good level of knowledge towards COVID-19 and its preventive measures [13]. The study is also corroborating with findings from a Systematic Review conducted during 2020 and a study conducted in southern

Ethiopia indicated for 61.78% and 63.51% of the participants, respectively had good knowledge towards COVID-19 preventive measures [1, 18].

In the knowledge assessment even though the majority heard about COVID-19 about 38.7% did not believe the existence of the disease. Moreover, during the study period, the majority (86.6%) agreed that COVID-19 is a killer and preventable disease, whereas, 8.2% disagreed for its severity. About 55.1%, 44.9%, and 82.2% claimed for the responsibilities of the government, community, and individuals respectively in implementing the COVID-19 preventive measures. However, **32.2%** of the study participants have favorable attitudes towards COVID-19 Preventive measures. This is less when compared with results in a survey of selected African countries (Egypt and Nigeria and southern Ethiopia) which showed most of the respondents (68.9%) and 54.5% had a positive and favorable attitude towards the protective measures being advised by the WHO or their local health authorities [1, 13]. In the same manner, 72.39% participants had favorable attitudes about Coronavirus in a systematic review conducted in Ethiopia [18].

In this study, 85.4% of the study participants practiced at least one of the COVID-19 Preventive measures endorsed by the government. About 3.12% did nothing to prevent COVID-19. The overall level of adherence to the implementation of COVID-19 preventive measure was 8.3%. This finding supports the findings from Southern Ethiopia which indicated that about 12.3% adhered to the recommended COVID-19 preventive measures [1]. However, it is far different from the study conducted in North Shoa and Gondar city at the binging of the pandemic that showed the overall adherence of the community to COVID-19 mitigation measures of 44.1% and 51.01%, respectively [2, 7]. The difference might be due to the fact that the current study was conducted in both urban and rural areas after people's give-up and loss of hope. While the later was researched in urban and just during the occurrences of few cases in Ethiopia.

In the final model, as age increases the odds of good level of adherence to COVID-19 preventive measures also increase. This supports the notion of the older age of 64 years or greater had higher odds of having knowledge on the prevention methods of COVID-19 for about 11 times higher compared to ages below 18 years old [19]. Also, in another study, participants in the ≥65 year's age group were 2.72 times more likely to have adherence to COVID-19 preventive measures as compared to the 35 and less year's age group [20].

The odds of good level of adherences to COVID-19 preventive measures increases with level of education. This supports the finding from Dessie and Kombolcha for the participants who were attending high-level education were 60% times more likely to have adherence to COVID-19 preventive measures compared with those who were unable to read and write [19]. In another study, a higher level of education was associated with better preventive behaviors [15].

Being a merchant was less likely to have a good adherence to COVID-19 preventive measures. This is supported by the results from the qualitative methods in which most of the time merchants going from place to place are not practicing the preventive measures. One of the discussants said,

*In our village, for example, women selling onion, tomato and other daily consumables working at "Gulit" may not practice physical distancing, hand washing, stay at home and almost all preventive measures because they are working to win their daily bread for their family.*

**A 31 years female**

The study also showed that the odds of having a good adherence to COVID-19 preventive measures is lower among participants with poor level of knowledge on COVID-19. As knowledge is the result of awareness based on obtaining appropriate information, it is supported by a study conducted elsewhere [7]. This finding is congruent with a study conducted in the Netherlands [21] which showed high information seeking behavior is associated with a good adherence to COVID-19 Preventive measures. This might be due to the fact that if the population had prior information about the utilization and advantage of the prevention measures [21].

In this study, the attitude of the respondents towards COVID-19 preventive measures failed to be significant in the multivariate analysis. However, in several studies those respondents having favorable attitude towards COVID-19 preventive measures were more likely to adhere to the mitigation measures than their counterparts [21]. The possible explanation might be that the respondents who had a favorable attitude towards COVID-19 preventive measures might trust the science of mitigation measures and comply with the instructions of these guidelines [21]. This is supported by the results from qualitative methods in which, social factors such as negative attitude towards those people practicing the prevention measures were the main factors for not adhering to the prevention measures. People consider those wearing facemask and using sanitizer as *foreigners and those who fear dea*th. Hence, people do not practice the prevention measures not to be labeled as such and not to be unique. Moreover, the key informant in Borena Health Office indicated that "stigma is also among the factors that affected use of prevention method." Moreover, some people claimed that there is no Coronavirus in the country." The key informant further states,

*the government only propagates the existence of the disease to get the foreign aid. Even some of the health professional claimed that it is only to get perdiem. There are also people who say we are protected by our almighty God. So, no need of use of prevention method.*

## Limitations of the study

Firstly, due to the cross-sectional nature of the study design, it might be difficult to ascertain the cause effect relationship between the study variables. Secondly, social desirability bias might be introduced despite their poor actual implementation. Thirdly, the tool used in this study was developed by the research team based on the context and not previously validated and the reliability was checked using Cronbach's alpha.

## Strength of the study

Through this community-based survey, it was possible to conduct a face-to-face interview and observation with maximum precaution than a simple telephone survey as others during the pandemic to evaluate the real response and adherence of the community to the mitigation measures. This study was conducted in a highly spreading time of the pandemic being an input for the government and others actors to intervene.

## Conclusions

This study found that highest awareness level (**91.6%**), moderate level of knowledge (**58.4%**) and low favorable attitudes **(32.2%)** towards COVID-19 Preventive measures were observed. About 85.4% practiced at least one of the preventive measures endorsed by the government. The overall level of adherence to COVID-19 preventive measure was very low **(8.3%)**. Age group, level of education, having poor level of knowledge on COVID-19 [AOR, were factors

associated with level of adherence to COVID-19 preventive measures]. In qualitative method, political context, unemployment nature of livelihoods, and necessity of social events were mentioned as a reason for the poor adherences to COVID-19 preventive measures.

Based on the findings of the current study it is possible to recommend that, activities to increase the knowledge, attitude and adherences to COVID-19 and its preventive measures through appropriate information outlets such as radio on continuous bases. Much work is needed from the concerned bodies like the government and/or the health sectors in improving the adherence of the community to the recommended safety measures of COVID-19. More-over, it is crucial to enforce the health regulations towards the preventive measures endorsed by the government. Preparation and dissemination of teaching aids prepared in local languages considering the socio-demographic, political and cultural factors are crucial to improve the community's adherence to COVID-19 preventive measures. The government officials have to consider some of their actions including meetings, gatherings at different levels since it might pass wrong message to the community like believing that COID-19 does not exist. Legal enforcement for COVID-19 prevention has to be revitalized as well as possibility with serious precaution to be followed and implemented.

## Supporting information

**S1 Data.**
(ZIP)

**S1 Questionnaire.**
(DOCX)

## Acknowledgments

The study team would like to acknowledge ORHB for facilitating the research. We also, greatly acknowledge the study participants for their cooperation. Staff members of ORHB and the data collectors deserve recognition for their facilitation and contribution.

## Author Contributions

**Conceptualization:** Sileshi Garoma Abeya, Dereje Duguma Gemeda, Mirgisa Kaba Sarbesa, Eliyas Yosuf Yesuf.

**Data curation:** Sileshi Garoma Abeya, Sagni Bobo Barkesa, Asebe Feyera Tolera.

**Formal analysis:** Sileshi Garoma Abeya, Fekadu Yadeta Muleta, Mengistu Bekele Hurisa.

**Investigation:** Sileshi Garoma Abeya, Sagni Bobo Barkesa, Dereje Duguma Gemeda.

**Methodology:** Sileshi Garoma Abeya, Sagni Bobo Barkesa, Chala Gari Sadi, Fekadu Yadeta Muleta, Asebe Feyera Tolera, Mirgisa Kaba Sarbesa, Asebe Amenu Tufa.

**Project administration:** Sileshi Garoma Abeya, Chala Gari Sadi, Seada Ahmed Mohammed, Endale Bacha Wako, Dereje Abdena Bayisa.

**Supervision:** Sileshi Garoma Abeya, Sagni Bobo Barkesa, Chala Gari Sadi, Dereje Duguma Gemeda, Fekadu Yadeta Muleta, Asebe Feyera Tolera, Dashe Negewo Ayana, Mengistu Bekele Hurisa, Dereje Abdena Bayisa.

**Validation:** Sileshi Garoma Abeya, Mirgisa Kaba Sarbesa.

**Visualization:** Sileshi Garoma Abeya.

**Writing – original draft:** Sileshi Garoma Abeya, Dashe Negewo Ayana, Mengistu Bekele Hurisa, Mirgisa Kaba Sarbesa, Eliyas Yosuf Yesuf, Asebe Amenu Tufa.

**Writing – review & editing:** Sileshi Garoma Abeya, Sagni Bobo Barkesa, Dereje Duguma Gemeda, Eliyas Yosuf Yesuf, Asebe Amenu Tufa.

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
