## [Decision Letter · Decision Letter 0]

22 Jun 2021

PONE-D-21-15954

Adherence to COVID-19 preventive measures and associated factors in Oromia regional state of Ethiopia

PLOS ONE

Dear Dr. Abeya,

Thank you for submitting your manuscript to PLOS ONE. After careful consideration, we feel that it has merit but does not fully meet PLOS ONE’s publication criteria as it currently stands. Therefore, we invite you to submit a revised version of the manuscript that addresses the points raised during the review process.

Please only pay your attentions to the errors or clear statements.Please submit your revised manuscript by Aug 06 2021 11:59PM. If you will need more time than this to complete your revisions, please reply to this message or contact the journal office at plosone@plos.org. Please include the following items when submitting your revised manuscript:
A rebuttal letter that responds to each point raised by the academic editor and reviewer(s). You should upload this letter as a separate file labeled 'Response to Reviewers'.A marked-up copy of your manuscript that highlights changes made to the original version. You should upload this as a separate file labeled 'Revised Manuscript with Track Changes'.An unmarked version of your revised paper without tracked changes. You should upload this as a separate file labeled 'Manuscript'.

We look forward to receiving your revised manuscript.

Kind regards,

Jianguo Wang, PhD

Academic Editor

PLOS ONE

Journal Requirements:

4. Please include a copy of the interview guide used in the study, in both the original language and English, as Supporting Information, or include a citation if it has been published previously."

5. Thank you for stating in the text of your manuscript "Informed verbal and written consents were obtained from residents, service users, service providers and policymakers prior to their enrolment in the study" and "Ethical approval was obtained from the ethical review board of Oromia Regional State Health Bureau. Permission letters were secured from Regional and Zonal Health Offices and shared with the randomly selected health care facilities and community administrators. Assent for less than 18 years and verbal consent was obtained from participants." Please also add this information to your ethics statement in the online submission form.

6. Please include your tables as part of your main manuscript and remove the individual files. Please note that supplementary tables (should remain/ be uploaded) as separate "supporting information" files

7. Thank you for stating the following in the Acknowledgments Section of your manuscript:

'The study team would like to acknowledge ORHB for funding the research.'

'This research was conducted as a routine ministry activities during COVID-19

campaign with out a dedicated fund'

8. We note that you have indicated that data from this study are available upon request. PLOS only allows data to be available upon request if there are legal or ethical restrictions on sharing data publicly. For information on unacceptable data access restrictions, please see http://journals.plos.org/plosone/s/data-availability#loc-unacceptable-data-access-restrictions.

9. Please upload a copy of Figure 12, to which you refer in your text on page 19. If the figure is no longer to be included as part of the submission please remove all reference to it within the text.

Additional Editor Comments (if provided):

Reviewers' comments:

Reviewer's Responses to Questions

**Comments to the Author**

1. Is the manuscript technically sound, and do the data support the conclusions?

Reviewer #1: Yes

Reviewer #2: Yes

2. Has the statistical analysis been performed appropriately and rigorously? 

Reviewer #1: Yes

Reviewer #2: Yes

3. Have the authors made all data underlying the findings in their manuscript fully available?

Reviewer #1: Yes

Reviewer #2: No

4. Is the manuscript presented in an intelligible fashion and written in standard English?

Reviewer #1: Yes

Reviewer #2: Yes

5. Review Comments to the Author

Reviewer #1: The manuscript is good however the following sections need to be revised to make it more scientific.

a. The introduction section of the manuscript seems long. It is better to reduce it focusing on the main issue, rationale and objective of the study giving the references of national and international scenarios.

b. Methodology part is too long, needs to describe briefly.

I have attached the feedback file for further section wise comments and feedback with yellow highlights.

Reviewer #2: Manuscript is written in good and standard English, with a few typos and grammatic errors that can be corrected during final proof reading. The methods sections is replicable and in specifically the Statistical analysis was done appropriately with regressions well explained and final models tested for normality and collinearity. Additionally, it is technically sound, with all conclusions being developed from the results. A few things need to improved as indicated in the attached file.

In conclusion, the paper is acceptable.

6. PLOS authors have the option to publish the peer review history of their article (what does this mean?). If published, this will include your full peer review and any attached files.

Reviewer #1: **Yes: **Dr. Kapil Amgain

Reviewer #2: No

---

## [Author Response · Author response to Decision Letter 0]

17 Jul 2021

Author's response to reviews:

Reviewer #1 

We thank the reviewer as s/he has made us to locate issues by providing both general as well as specific comments that eventually strengthen the manuscript. Accordingly, we have tried to accommodate the comments.

a. The introduction section of the manuscript seems long. It is better to reduce it focusing on the main issue, rationale and objective of the study giving the references of national and international scenarios.

• This comment is well accepted. Hence, we have tried to make it short giving more attention to rationale and objective of the study giving the references.

b. Methodology part is too long, needs to describe briefly.

• The comment is accepted and correction is made accordingly

c. The attached file for further section wise comments and feedback marked with yellow highlights are addressed.

Background

• It is not corona virus, it is Coronavirus.

Yes, we have accepted the comments and corrected it

• Check this sentence and rewrite it to make sense.

Yes, we have corrected the sentence

• No need to spell out again, Just COVID-19 is enough

Yes, we have corrected 

Methods and Materials

• How is it possible to conduct a community based study in COVID-19 pandemic and lock down throughout the globe?

The concern is right. As the study was conducted between September 2020 to March 2021, the lockdown was not practical and declared in Ethiopia 

• How did you conduct the survey? Did you use questionnaire? If yes please provide the questionnaire.

Ok! The questionnaire is attaches as a supplementary file

• Why did you use both KII and FGD for qualitative data collection? why not one?

The KIIs were conducted to know the observation and actions taken by the health managers at different level and FGDs were held to explore the opinion and perception of the community. So the participants and discussants were different to explore comprehensive actions, opinion and perceptions

• Disclose for whom you have use KII and FGD

We have already mentioned the participants of KIIs and discussants of FGDs in the document

Ethics Consideration

• The less than 18 years of people are in exclusion criteria of your study, why it is necessary to take assent from them??Clarify it

Yes, the comment is accepted and correction is made on the document

Results

• is it majority? It is better to say majority more than 75% only

We have accommodated the comments 

Reviewer #2 

We would like to thank also the reviewer for sharing his/her view and valuable comments. We the authors briefly have a discussion on the comments and tried to incorporate accordingly. The point by point corrections and responses are as follow:

Methods

• Sample size determination of the quantitative data is scientifically drawn. However, cite/ reference the formula.

The comment is accepted and citation is made to the formula

Results

• The estimated annual income of the respondents can be also be put in US dollars to give and an easier understanding to the international community

Ok! The comment is accepted and we have changed in to USD

• What is the difference between rural and urban communities with regard to the knowledge of COVID-19 including its preventive measures? Can author show proportions for rural and urban differently?

Yes! Among urban/town residents 63.0% have good level of knowledge compared to rural/wored having 52.3% which statistically significant (X2 = 30.8, P< 0.001). This statement is added to the main document

• When pointing out results for factors associated with adherence, the authors state an explanation of how the results were got (methods). I would suggest that that part removed.

Well- the paragraph is removed as it is mentioned in the method section

• Check out this statement “Accordingly, compared to study participants who attended Colleges and above, being Illiterate [AOR, 38; 95% CI: 0.15, 0.93]”. Here you need to correct that AOR. It seems incorrect.

Yes it was a typo error and corrected as the AOR= 0.38

• Include AOR value in this statement “Being merchant were less (AOR; 95% CI: 0.29, 0.96]”

Yes the results of AOR is included

• Include percentage sign in this statement “…above knowledge related variables and accordingly 1606 (58.4) have good level of knowledge”

Yes it was a typo error and corrected to 58.4%

Discussions

• Cite this stamen “This may be because the developing countries use social media less than developed countries and minimize disruption caused by the corona virus”

The comment is right, but the statement was the perception of the authors based on theoretical and conceptual knowledge and not based on empirical findings

• What is the implication of the knowledge level of 58%

The 58% and relatively lower level that have implications on attitudes and adherence of COVID-19 preventive measures in the study area

• Please share why you think this study had lower favorable attitudes towards COVID-19 Preventive measures as compared to other studies, including that in Ethiopia. In addition, show the implication of this finding.

The lower favorable attitudes towards COVID-19 Preventive measures might be attributed to the political situation in the region as it was explicitly mentioned in the results of qualitative approach

---

## [Decision Letter · Decision Letter 1]

31 Aug 2021

Adherence to COVID-19 preventive measures and associated factors in Oromia regional state of Ethiopia

PONE-D-21-15954R1

Dear Dr. Abeya,

We’re pleased to inform you that your manuscript has been judged scientifically suitable for publication and will be formally accepted for publication once it meets all outstanding technical requirements.

Kind regards,

Jianguo Wang, PhD

Academic Editor

PLOS ONE

Additional Editor Comments (optional):

Reviewers' comments:

Reviewer's Responses to Questions

**Comments to the Author**

1. If the authors have adequately addressed your comments raised in a previous round of review and you feel that this manuscript is now acceptable for publication, you may indicate that here to bypass the “Comments to the Author” section, enter your conflict of interest statement in the “Confidential to Editor” section, and submit your "Accept" recommendation.

Reviewer #1: All comments have been addressed

2. Is the manuscript technically sound, and do the data support the conclusions?

Reviewer #1: Yes

3. Has the statistical analysis been performed appropriately and rigorously? 

Reviewer #1: Yes

4. Have the authors made all data underlying the findings in their manuscript fully available?

Reviewer #1: Yes

5. Is the manuscript presented in an intelligible fashion and written in standard English?

Reviewer #1: Yes

6. Review Comments to the Author

Reviewer #1: Dear PLOS ONE Team,

Thank you for providing me with the review opportunity.

I have already sent my feedback to the manuscript in email, as I was unable to upload my feedback in the PLOS editorial manager portal.

If you missed my email then here is my feedback, kindly proceed forward on behalf of me.

My Comments and Feedback:

As this is the second round review of the manuscript entitled with " "Adherence to COVID-19 preventive measures and associated factors in Oromia regional state of Ethiopia."

I went through the author's revision of my comments, and found that the author has revised it as per my review feedback and now it is better than previously submitted one.

You can proceed towards publication.

Thank you for your cooperation

Dear PLOS ONE Team,

Thank you for providing me with the review opportunity.

I have already sent my feedback to the manuscript in email, as I was unable to upload my feedback in the PLOS editorial manager portal.

If you missed my email then here is my feedback, kindly proceed forward on behalf of me.

My Comments and Feedback:

As this is the second round review of the manuscript entitled with " "Adherence to COVID-19 preventive measures and associated factors in Oromia regional state of Ethiopia."

I went through the author's revision of my comments, and found that the author has revised it as per my review feedback and now it is better than previously submitted one.

You can proceed towards publication.

Thank you for your cooperation

7. PLOS authors have the option to publish the peer review history of their article (what does this mean?). If published, this will include your full peer review and any attached files.

Reviewer #1: No

---

## [Editor Report · Acceptance letter]

6 Sep 2021

PONE-D-21-15954R1 

Adherence to COVID-19 preventive measures and associated factors in Oromia regional state of Ethiopia 

Dear Dr. Abeya:

I'm pleased to inform you that your manuscript has been deemed suitable for publication in PLOS ONE. Congratulations! Your manuscript is now with our production department. 

Kind regards, 

on behalf of

Dr. Jianguo Wang 

Academic Editor

PLOS ONE